# Volcanic Tuff as Secondary Raw Material in the Production of Clay Bricks

**DOI:** 10.3390/ma14226872

**Published:** 2021-11-15

**Authors:** Nicoleta Cobîrzan, Gyorgy Thalmaier, Anca-Andreea Balog, Horia Constantinescu, Andrei Ceclan, Mircea Nasui

**Affiliations:** 1Department of Civil Engineering and Management, Faculty of Civil Engineering, Technical University of Cluj-Napoca, 25 Baritiu Street, 400624 Cluj-Napoca, Romania; Nicoleta.Cobarzan@ccm.utcluj.ro; 2Department of Materials Science and Engineering, Faculty of Materials and Environmental Engineering, Technical University of Cluj-Napoca, 103 Muncii blv., 400624 Cluj-Napoca, Romania; 3Department of Structures, Faculty of Civil Engineering, Technical University of Cluj-Napoca, 25 Baritiu Street, 400624 Cluj-Napoca, Romania; Anca.Balog@dst.utcluj.ro (A.-A.B.); Horia.Constantinescu@dst.utcluj.ro (H.C.); 4Department of Electronics and Measurements, Faculty of Electrical Engineering, Technical University of Cluj-Napoca, 26 Baritiu Street, 400624 Cluj-Napoca, Romania; Andrei.Ceclan@ethm.utcluj.ro; 5Department of Physics and Chemistry, Faculty of Materials and Environmental Engineering, Technical University of Cluj-Napoca, 103 Muncii blv., 400624 Cluj-Napoca, Romania; mircea.nasui@chem.utcluj.ro

**Keywords:** brick, clay, tuff, raw materials, waste

## Abstract

The present work examines an innovative manufacturing technique for fired clay bricks, using tuff as a secondary raw material. Samples were made of clay and tuff (0–30 wt.%) fired at 900 to 1100 °C. The chemical and mineralogical compositions and physical and thermal analyses of raw materials were investigated by using SEM-EDS, RX and DTA-TG curves. The samples were analysed from the mineralogical, technological and mechanical points of view. The result show that the tuff’s presence in the clay mixtures considerably reduced the shrinkage of the product during the firing process, and the manufactured samples were of excellent quality. The compressive strength of the bricks varied from 5–35.3MPa, being influenced by the tuff content, clay matrix properties and firing temperatures. Finally, the heat demand for increasing the temperature from room to the firing temperature of the sample with 10% tuff content was 22%.

## 1. Introduction

The manufacturing of building materials and products is responsible for about 11% of CO_2_ emission at the global level [1], impacting global warming and constantly contributing to climate change. Clay brick is one of the most used building materials in the world, in the construction of structural walls, arches and vaults and for infill panels, in the case of framed structures. The manufacturing process of such products is energy intensive, requiring raw materials preparation and the extrusion, drying and firing of green bricks at temperatures up to 950 °C, which are required for mineral phases’ transformation to occur. The total CO_2_ emissions generated during the production phase is influenced both by the fuel consumption used in all stages (the transport of raw materials from the quarry to the fabrication site, drying, firing etc.) and decomposition of minerals during the firing process. Even though the energy used for brick production has been reduced since 1990 [2], the role of industry in transitioning to climate neutrality [3] may be significant.

The potential in reducing the environmental impact and the depletion of raw materials may be achieved by using secondary materials and renewable energy systems to save conventional energy at the level of the brick industry and, consequently, in efficiently sustaining the growth of the global economy. The adoption of the circular economy model [3,4,5] is necessary in this approach, including innovative techniques to “close the loop” of products through their reuse or recycling at the end of their life cycles.

The recycling processes and transformation of waste into secondary raw material can also be energy-intensive, requiring the selective collecting and removal of waste from sites, and the treatment (drying, grinding at optimum grain size etc.) and preparation as secondary raw materials for applications in different industries according to their technical characteristics.

EU directive 2008/98/EC [6] established a general framework for waste management, recovery and recycling and reduced GHG emissions. The EU aims to develop a common framework for the construction-materials market [7] to maximize the reuse of waste materials, with a clear impact in industry and building decarbonization.

Due to their strength, endurance, natural beauty and the ease of their manufacturing, zeolitic tuffs are an important building material in architectural ornaments but also as a building material for everyday housing. Tuffs were used in the Pannonian–Carpathian region from pre-roman age as a building material, and are found in fortresses, castles and churches, as well as in houses, since it is a readily available local resource [8]. As a result, there are a considerable number of buildings in the Transylvanian and Pannonian basins made of this material that are in end-of-life status. The aim of this research was to study an alternative to landfills for the demolition waste generated by these buildings, and, also, to find a recovery solution for waste generated during the cutting and extraction of stone. 

The uses of tuff as binder or aggregate in buildings materials has been studied by many researchers in the last decades. Some analyses have provided evidence of the pozzolanic properties of volcanic tuff [9] and the possibility for its use as an admixture in mortar or concrete composition [10,11,12,13,14,15], as lightweight concrete blocks [16,17] or in asphalt mixes [18], while others have shown their potential use as masonry units [19] in masonry structural elements.

Tuff was used as an additive in clay matrices, in various percentages, by weight, as a humidity control [20]. The role of natural zeolites content in a clay matrix was investigated by the authors [21], showing that the thermal conductivity of fired bricks had been reduced. According to [22], bricks with the addition of zeolites (30 wt.%) decreased its compressive strength, but not below the minimum value imposed by standards.

Most of the papers dealing with the use of volcanic tuff in the building industry concern the incorporation of tuffs in cement or geopolymers [23] for use as building materials. The literature reveals a limited number of papers concerned with the incorporation of tuff in fired clay bricks [20,21,22].

Dej Tuff is the most prominent acidic tuff of the whole Carpathian–Pannonian area, and its geographical localization is the northwestern and northern border of the Transylvanian Basin. This tuff has a lower Badenian age [8,24,25] and is found in layers of variable thickness, from a few meters up to 30–40 m.

The aim of this study was to determine the maximum percentage of tuff that can be added to a clay mixture such that the physical–mechanical characteristics of the final product comply with the technical requirements provided by design codes.

In order, the paper presents the relevant background; details of the clay mixtures themselves; the procedure of samples’ preparation and the adopted methodology; and then highlights a microstructural analysis of the raw materials and a macrostructural characterization of both green and fired samples. In the last section we present final remarks, conclusions and proposals for further work.

## 2. Materials and Methods

### 2.1. Specimens Preparation

The boundary system for the brick samples was settled according to (Figure 1) to evaluate their potential for reducing environmental impact by substituting the clay with volcanic tuff. The tuffs were obtained from a local source (Transylvanian basin Romania), dried and milled.

The samples were a mixture of yellow/grey clay (70:30% by mass, containing mainly quartz, kaolinite, biotite and calcite/kaolinite, quartz, dolomite, calcite [26]) and volcanic tuff in 0% (S1), 5% (S2), 10% (S3), 20% (S4) and 30% (S5) of total mass. All raw materials were dried before use at 90 °C/1h. The specimens had a cylindrical shape with a diameter and height of 18 mm, pressed (40 N/mm^2^) with a hydraulic press and then fired in an electrical oven at temperatures of 900 °C, 1000 °C and 1100 °C for each mixture. Ceramic materials were shaped by the dry pressing method to eliminate structural defects caused by deformation and shrinkage during the drying and firing processes, but also to reduce the consumption required for sample drying in the preliminary stage.

The firing temperature was gradually increased by 2°/min from room temperature up to a temperature of 600 °C and then, by 5 °C/min, to the final temperature. The samples were held at the final temperature for 2 h, and the heat treatments were conducted in air, as in [27]. Before and after firing, the samples were analysed for their structural, mechanical, and physical properties.

### 2.2. Analytical Methods

The morphologies of the particles and the local chemical composition were determined by SEM–EDS on a Jeol 5600 LV scanning electron microscope. Uncoated, fresh fractures of the fired samples were also subjected to SEM-EDS analysis. The EDS analysis used the ZAF correction standards, implemented in the AZTEC 4.0 software. The particle-size distribution of the raw materials was determined by intrusion mercury porosimetry (Pascal 140). By this method, the size range of the particles is calculated using the pressure needed to break the forces binding the particles together in the agglomerates in the low-pressure regime, according to the Mayer–Stowe theory. The thermal behaviour during the heating process of the press-ready mixture was evaluated using a combined DTA-TG analysis at up to 1000 °C, in air, with a heating rate of 10 °C/min (modified MOM Hungary). X-ray diffractions were recorded on an INEL–Equinox 3000 diffractometer using Co Ka1 radiation (λ = 0.17903 nm). Al samples were ground manually to a fine powder and treated with HF to decrease the SiO_2_ content.

Compressive strength was measured on a Controls Advantest 9 hydraulic press with a load rate of 0.2 MPa/s on samples having a d/h ratio of ~1. The apparent densities of the samples were determined according to SR EN 772-13:2001 [28].

The total shrinkages of the samples were calculated from the samples’ dimensions measured using a calliper as (Di−Df) × 100/Di, where Di is a sample’s initial diameter and Df its diameter after firing. Sample colour was estimated from sample photos as RGB hexadecimal values.

## 3. Results and Discussion

Scanning electron microscopy analysis of the raw materials was performed by using topographic contrast, due to the different distances travelled and the number of electrons emitted from the surface of the sample. All visible particles were agglomerated into coarser, irregular-shaped structures (Figure 2a–c).

The raw materials were analysed using the EDS probe (Table 1). 

From the EDS results it was found that all three raw materials had a high content of SiO_2_ (61–74.6%), with significant amounts of Al_2_O_3_, Fe_2_O_3_ and MgO and traces of Na_2_O. It may be observed that all raw materials have similar chemical compositions. From a compositional point of view, their alumina contents placed the materials in the category of semi-acidic clays with a significant chromophore oxide (iron oxide) content. An important aspect of the used tuff is its high fraction of glassy content, as suggested by the presence of amorphous, wide peak in the X-ray pattern.

Particle-size analyses of raw materials are presented in Table 2.

The used clays can be considered “dusty”, containing significant amounts of fine particles (<5 µm) and containing almost no particles with size >63 µm. As is well known from classical powder metallurgy, very fine particles cannot be conveniently formed into dense bodies, since extremely narrow channels inhibit air from being evacuated from the particles. In the present case, the resulting porosity can be of benefit, by reducing the future bricks’ mass, provided their compression and green strengths can be maintained at acceptable limits. The forming pressure was chosen as the lowest pressure that gave sufficient green strength for the samples to be easily handled.

As can be seen in the compression curves of the mixtures (Figure 3), increasing pressure did not significantly increase density, a parameter that strongly influences the final mechanical properties on parts manufactured by powder metallurgy from the samples. These naturally agglomerated powder mixtures can be easily formed, by pressing, into a desired shape and have decent green strengths. This fact permits the easy handling of the samples; no spring back-related defects or cracks were visible in the samples, even after 48h of rest.

The X-ray diffractions on the samples fired at 900 °C are presented in Figure 4a. The presented peaks are indexed as originating from the quartz (PDF file 85-0865), feldspar (PDF file 70-1862) and hematite (PDF file 89-8104) Some traces of muscovite (PDF file 46-1409) were also present in samples with 5% tuff. The other chemical elements probably formed an amorphous matrix that bound everything together. Due to the high intensity of the quartz peaks the other phases with less intense peaks were difficult to identify. The macroscopically homogeneous matrix analysed on a mesoscopic level could be described as composed by a glassy matrix holding together the different crystalline phases (quartz, hematite) and containing the embedded porosity, similar to a particle-reinforced composite material.

During the firing process these raw materials underwent a series of transformations and decompositions that influenced the final properties of the samples. A DTA-TG curve is presented in Figure 4c. Although no water was added, the natural moisture from the samples evaporated in the low temperature regime and accounted for a mass loss of approximative 4%.

A second endothermic effect was present in the 250–500 °C and caused by the loss of the strongly associated water. At approximately 500 °C another event overlapped with the bonded water loss, the dehydroxylation of kaolinite which became metakaolinite. At temperatures over 700 °C carbonate decomposition occurs.

### 3.1. Density

The densities of the samples were determined before and after firing in the oven at 900 °C (Figure 5). By adding tuff to the mixture, the samples’ densities were slightly reduced from 1.71 g/cm^3^ (reference sample) to 1.51 g/cm^3^ (S5) (Table 3). After the firing at a temperature of 900 °C, the samples densities were decreased by 10.9% (reference sample), 8.6% (S2), 9.8% (S3), 8.7 (S4) with 4.4% (S5), as compared with the samples’ densities measured before firing (Figure 5).

The densities of the samples after firing at the different temperatures (900 °C, 1000 °C and 1100 °C) are shown in Figure 5. The densities of the samples S1 and S2 a decreased as firing temperature increased due to continuous mass loss. By increasing the tuff content, sample densities decreased due to the lower density of the used tuff (1.36 g/cm^3^) compared with the other raw materials. Increasing the tuff content also modified the chemical composition of the samples and thus their densifications.

The tuffs used were rich in zeolites, which are aluminium tectosilicates that act as tetrahedral carcasses of silicon oxides with the role of molecular sieves [29], ion exchangers, humidity regulators and in the release of “zeolitic water” at temperatures above 100 °C. The use of tuffs with high a content of zeolites can contribute to the better sintering of the entire mixture with the appearance of a glassy mass at temperatures above 1000 °C, which may justify the increase of mechanical strength.

Based on the results obtained, the ceramic materials were of high density, according to SR EN 771-1:2015 [30], which can be used in structural and non-structural masonry elements, if all the requirements imposed by design codes are fulfilled [31,32]. In the context of sustainability, the density of materials may also play an important role; at the building level, the reduced density of bricks as masonry units can lead to the reduction structural elements’ contributions and, consequently, the reduction of needed reinforcements in order for the future building to withstand the previsioned loads. The reduced need for materials directly impacts the carbon footprint of the building.

### 3.2. Shrinkage

One way of comparing the degree to which samples’ were sintered is their total shrinkage during the solid-stage sintering process [33]. At the initial stage, the dimensional variation is minimal; no significant shrinkage should be visible. In the second stage of sintering, considerable densification occurs, leading to important shrinkage. The ultimate goal of the sintering process, in the present study, was to obtain good compressive strength, yet, at the same time, to have minimal densification in order to preserve as much porosity as possible.

In Figure 6 the dimensional changes of the samples after the firing process are presented. In the graph, the results of firing at 1100 °C indicate a stronger, more intense sintering. In the ternary phase diagram of the K_2_O–Al_2_O_3_–SiO_2_ system [34] no liquid phase was formed in the samples fired up to 1000 °C. This is in accordance with the DTA analysis presented in Figure 4. In the first two heating regimes, the samples were subjected to solid-state sintering, so the mass transport in this case is rather limited and the shrinkage is low. The low sintering degree suggests the samples were in the end of the first stage of sintering (see Figure 7). At this stage, the dominant mass transport mechanism is surface diffusion. The samples mainly consist of small particles, so their specific surface areas are high. Even if the samples are in the initial stage of sintering, with no major impact on the porosity their mechanical strength is high. The linear shrinkage of samples increased as more tuff was added, but even so, it was considerably lower than in the case of the control sample (S0) (Figure 7). At temperatures < 1000 °C the shrinkage was less than 1% for all samples, lower, by a factor of three, than other authors findings [20]. The numerical value of the measured shrinkages is presented in Table 4.

In the secondary electron micrographs, it was evident that vitrification was more extensive as the tuff content increased. Additionally, the sintering necks (examples marked by the black arrows) were stronger as the temperature increased, also confirming stronger sintering. Increasing the temperature to 1100 °C, some liquid phase formed, and the liquid-phase sintering changed the microstructure; the initial particles were hard to see, having merged into a continuous matrix circumventing the pores. The shrinkage was high; 5.6% at 30% tuff, but even so it was 30% lower than in the case of the control sample.

The different sintering was also visible, macroscopically, in the samples by a colour change from reddish (#908579), at 900 °C, to a light-brownish appearance (#89786e) at 1000 °C, turning to grey (#454f5b) at 1100 °C. No significant colour variation was observed as a function of increasing tuff content.

### 3.3. Compressive Strength

The compressive strength of materials is the most important parameter of masonry units, according to which their usage in construction is established. The mechanical strength of the analysed samples was highly influenced by the tuff content and firing temperature. Sample S5, fired at a temperature of 900 °C, had the compressive strength of 5MPa, 68% lower than the reference sample (S1, Table 5). At 900 °C and 1000 °C the raw material particles were less sintered, due to their high SiO_2_ content, which sinters at higher temperature. When the firing temperature was increased to 1100 °C, compressive strength was increased due to the good bonding between the powder particles, achieved by better sintering. All samples fired at this temperature had compressive strengths between 25 MPa (S5) and 35 MPa (S3); compressive strength tended to be maximized for samples containing 10% tuff (Figure 8). The maximum value of compressive strength was reached because the highest density among all samples sintered at 1100 °C was obtained from this composition.

The high compressive strength, obtained at 1100 °C, allowed the incorporation of further waste material with a double role: to act as a space holder to further decrease density and, more importantly, to act as supplementary heat input to reduce the energy needed for firing the bricks [11]. Saw dust or other vegetal residues were successfully added to clay mixtures to act as pore-forming agents and to reduce heat input.

The obtained data suggests that the higher the tuff content, the lower the sample’s compressive strength. This can be linked to the quartz content, since the quartz undergoes a sudden, important volume change during the alpha–beta transformation. Although the heating rate was significantly reduced in the transformation temperature range, microcracks eventually appeared between the quartz and the surrounding matrix, especially in the case of larger SiO_2_ particles. These cracks can easily propagate through the glassy matrix. So, the importance of reducing compressive strength is not only concerned with increasing porosity but also with the matrix’s reduced capacity to block the propagation of cracks.

Similar trends of increased tuff percentage with decreased density and compressive strength have been observed in other studies [20,22,35] as evidenced in Table 6. Gencel et al. [22] and Cay et al. [35], from samples of very similar densities and manufactured by the semi-dry pressing process, obtained values of compressive strength, for samples with up to 30% tuff, greater than the minimum value required by national regulations.

Vakalova et al. [20] also observed a decreasing trend in the value of the compressive strength of clay–tuff samples manufactured by extrusion at temperatures up to 1050 °C, obtaining compressive strengths up to 23MPa. In their case, the shrinkage was extremely high, up to 14%; tuff had been added in their clay mixture as a humidity stabilizer with favourable effect or for reducing cracks originating from the drying process.

An increase in firing temperature can lead to increase of CO_2_ emissions and costs related to the firing process, however, this increase may be within acceptable limits. For a simplified approximation of the energy increase needed, we assumed a similar volumetric heat capacity for the clay minerals and the volcanic tuff [36,37]; the heat difference necessary for increasing the temperature from room temperature to the firing temperature depends on the sample’s mass. By using a high quantity of tuff with a low density (1.36 compared to 2.2 g/cm^3^ for the clays) the heat demand to increase the temperature of the sample S5 containing 30% tuff to 1100 °C was 116% of that needed for the sample S0 to reach 900 °C. In the case of sample S3, which had the best mechanical properties, the increase of heat demand for firing was approximative 22%. This can lead to the reduction of the environmental impact of these materials by allowing the manufacture of hollow bricks from a stronger material that allows a high void fraction. Using the above-mentioned simplification, we can conclude that the incorporation of a void fraction of over 20% can lead to more benign environmental impact, compared with the reference sample; yet, at the same time, the compressive strength should be above the minimal imposed value of 20 MPa.

## 4. Conclusions, Contributions, and Further Work

Herein, we have investigated the potential for using tuff as a secondary raw material in clay brick production. We analysed the optimal percentage of tuff that can be incorporated in a clay matrix. The results showed the following:✓The compressive strength of tuff–clay samples depended strongly on sintering temperature. When heated to 900 °C, the values of compressive strength were lower than 15.7 MPa for all samples (S2–S5). On the other hand, samples fired at 1100 °C presented higher compressive strengths, an increase of up to three-fold as compared with the reference bricks. This increase was due to the forming of stronger bonds between the particles. The increase in compression strength was also due to the formation of an important volume fraction of liquid that further accelerate the sintering process.✓The added volcanic tuff further reduced the firing shrinkage. The formation of the liquid phase accelerated the sintering and, therefore, the firing shrinkage. By correctly choosing the sintering conditions we managed to reduce shrinkage by almost one magnitude lower, when comparing sample S2 to the reference sample S1 at 1100 °C. In the other cases the shrinkage reduction was also significant, though the difference was lower as the tuff content increased.✓Increasing the firing temperature increased the embodied energy of the final product; however, by increasing the mechanical strength, one can add pore formers that improves thermal performance in energy-efficient buildings.

The present work showed that tuff can be successfully used as a secondary raw material in the fabrication of fired clay bricks, in ratios of up to 30%, thus, contributing to the circular economy and the EU’s zero-waste target. The results obtained in the present work will contribute to future research on the optimization of ceramic products based on tuff and other secondary raw materials, incorporated in a clay matrix as pore forming agent, in terms of thermal, mechanical and environmental performance.

## Figures and Tables

**Figure 1 materials-14-06872-f001:**
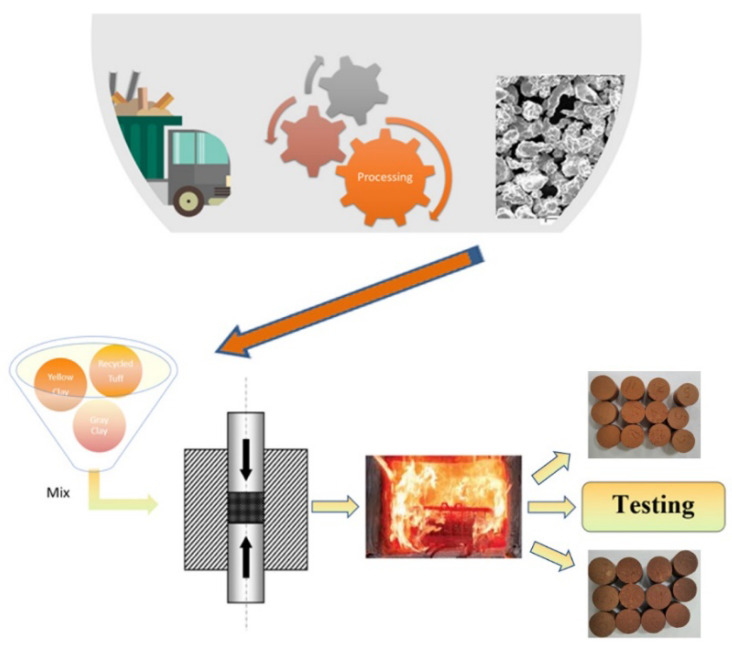
Specimen’s preparations.

**Figure 2 materials-14-06872-f002:**
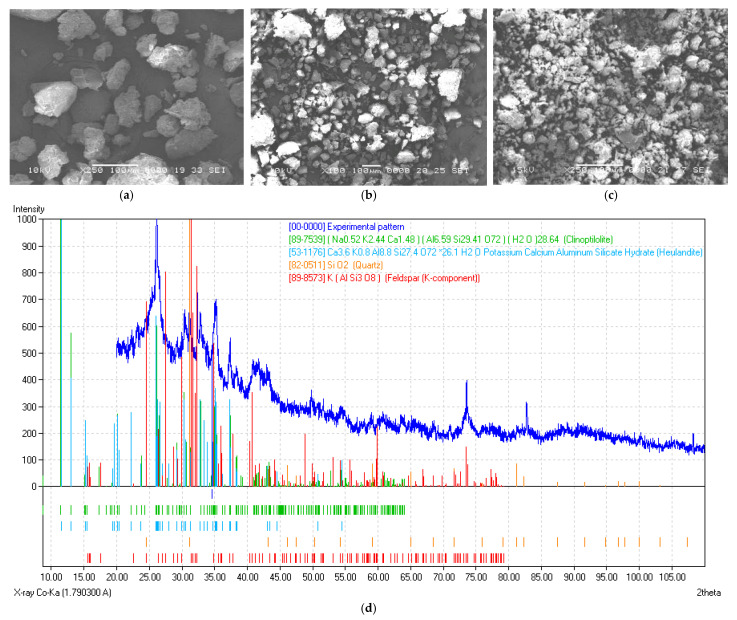
SEM secondary electron images of the initial powders: yellow clay (**a**); grey clay (**b**), tuff (**c**) and the X-ray diffraction pattern of the used tuff (**d**).

**Figure 3 materials-14-06872-f003:**
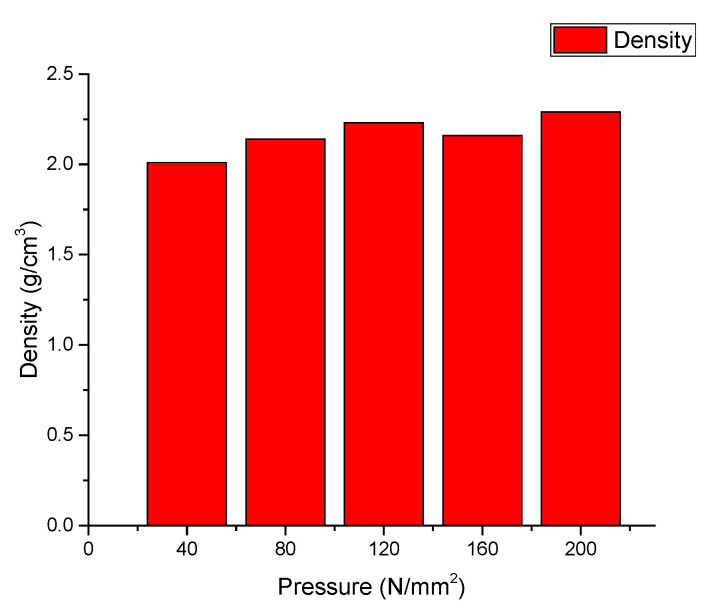
Sample compression curve of the mixture containing 10% tuff.

**Figure 4 materials-14-06872-f004:**
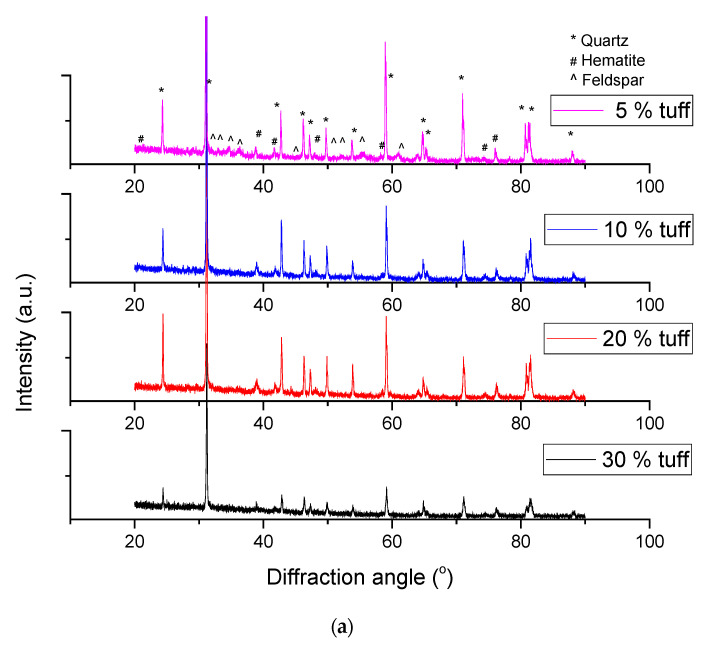
X-ray diffractions on samples fired at: (**a**) 900 °C, (**b**) 1100 °C and an example DTA-TG curve for the sample containing 10% tuff (**c**) (Δm—mass variation; Δt—temperature).

**Figure 5 materials-14-06872-f005:**
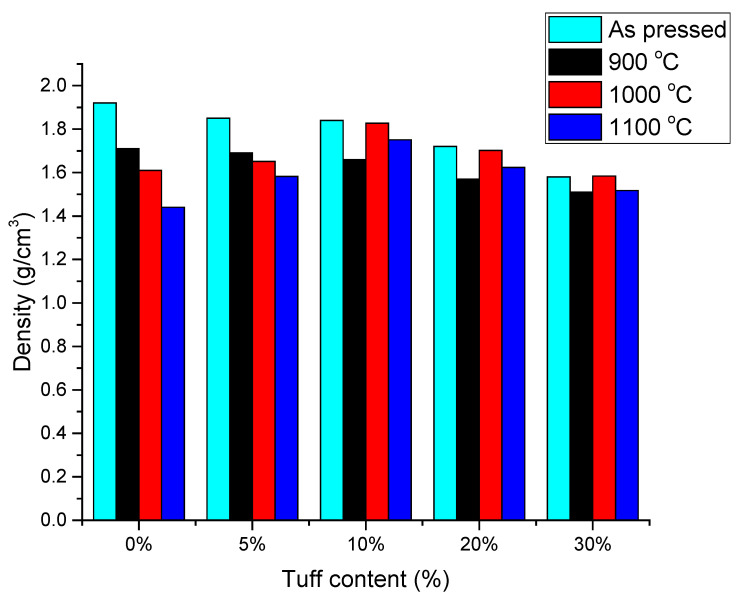
Effect of sintering temperature on the sample densities.

**Figure 6 materials-14-06872-f006:**
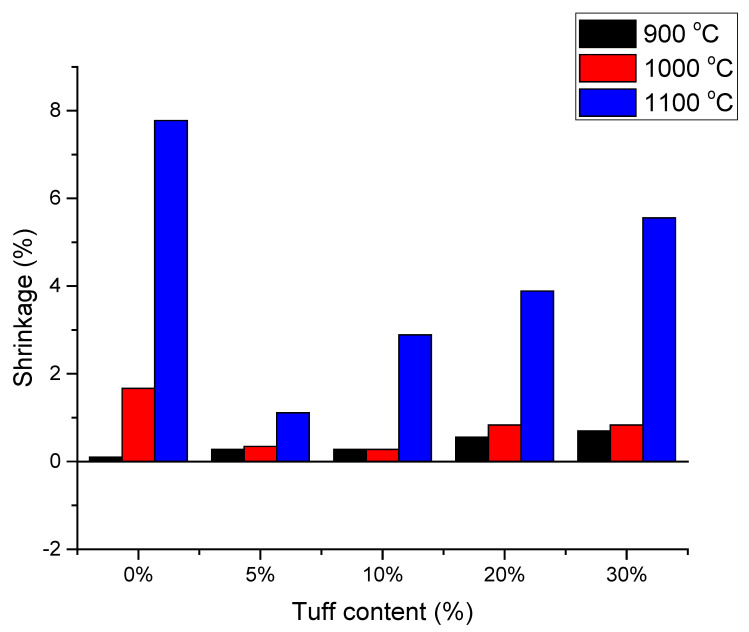
The influence of tuff content and temperature on shrinkage effect.

**Figure 7 materials-14-06872-f007:**
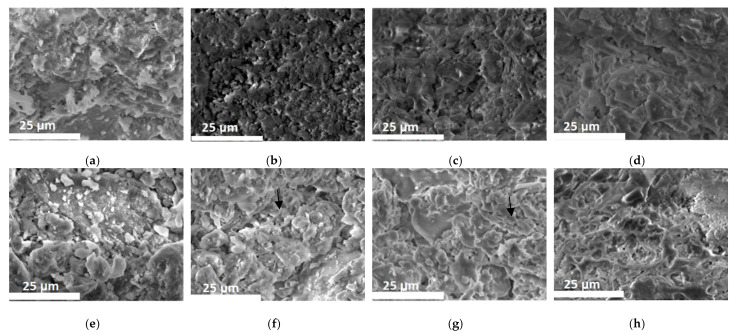
SEM: sample with 5% tuff content, unfired (**a**); fired at 900 °C (**b**); 1000 °C (**c**) and 1100 °C (**d**); samples with 30% tuff content unfired (**e**); fired at 900 °C (**f**); 1000 °C (**g**) and 1100 °C (**h**).

**Figure 8 materials-14-06872-f008:**
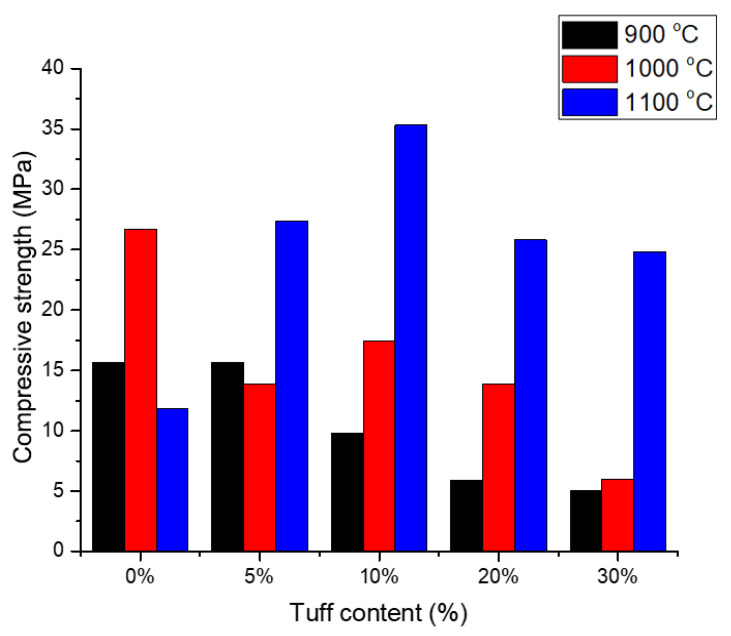
The values of compressive strength for the manufactured samples.

**Table 1 materials-14-06872-t001:** Elemental composition of raw materials (%).

Element	Yellow Clay	Grey Clay	Vulcanic Tuff
O	65.80	55.63	65.12
Na	0.00	0.25	0.21
Mg	1.61	1.45	0.54
Al	7.72	11.20	4.99
Si	18.76	22.28	25.81
K	1.85	5.37	1.64
Ca	1.58	2.01	1
Fe	2.67	1.81	0.68

**Table 2 materials-14-06872-t002:** Dimensional characteristics of clays and tuffs obtained by mercury porosimetry.

Particle Size Distribution	D10 *	D50 *	D90 *	<5 µm	>63 µm
Grey clay	2 µm	11 µm	35 µm	25%	<1%
Yellow clay	1.7 µm	9 µm	30 µm	32%	<1%
Volcanic Tuff	23 µm	30 µm	42 µm	0%	0%

* D10, D50, D90 signifies the point in the size distribution, up to and including which, 10%, 50% and 90% of the total volume of material in the sample is contained.

**Table 3 materials-14-06872-t003:** Measured densities of samples fired at 900 °C, 1000 °C and 1100 °C.

Tuff Content [%]	Fired at 900 °C [g/cm^3^]	Fired at 1000 °C [g/cm^3^]	Fired at 1100 °C [g/cm^3^]
0%	1.71	1.61	1.44
5%	1.69	1.65	1.58
10%	1.66	1.83	1.75
20%	1.57	1.70	1.62
30%	1.51	1.58	1.52

**Table 4 materials-14-06872-t004:** Shrinkage of brick samples fired at 900 °C, 1000 °C and 1100 °C.

Tuff Content (%)	Fired at 900 °C (%)	Fired at 1000 °C (%)	Fired at 1100 °C (%)
0%	0.1	1.7	7.8
5%	0.3	0.4	1.1
10%	0.3	0.3	2.9
20%	0.6	0.8	3.9
30%	0.7	0.8	5.6

**Table 5 materials-14-06872-t005:** Compressive strength of brick samples fired at 900 °C, 1000 °C and 1100 °C.

Tuff Content (%)	Fired at 900 °C (MPa)	Fired at 1000 °C (MPa)	Fired at 1100 °C (MPa)
0%	15.7 MPa	26.7 MPa	11.8 MPa
5%	15.7MPa	13.9 MPa	27.4 MPa
10%	9.8 MPa	17.4 MPa	35.3 MPa
20%	5.9MPa	13.9 MPa	25.8 MPa
30%	5 MPa	6 MPa	25 MPa

**Table 6 materials-14-06872-t006:** Comparison of some of the technical characteristics obtained in the present study with other similar studies.

Composition	Firing Temperature °C	Technical Characteristics		Reference
		Compressive Strength	Bulk Density	
		(MPa)	(g/cm^3^)	
70–100(wt.%) clay 0–30(wt.%) tuff	900	15.7–5	1.71–1.51	
	1000	26.7–6	1.61–1.58	This paper
	1100	35.3–11.8	1.44–1.52	
70–100(wt.%) clay 0–30 (wt.%) natural zeolite	1000 1050	32–24.2 33.8–23.1	-	[20]
70–100(wt.%) clay 0–30%(wt.%) natural zeolite	900	34.9–14.3	1.68–1.48	[22]
70–100(wt.%) red clay 0–30(wt.%) zeolite rock	900	34.9–14.3	1.68–1.48	[35]

## Data Availability

The data presented in this study could be available through the request from the corresponding author. The data are not publicly available due to privacy policies of the Technical University of Cluj-Napoca.

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
