# Peer review of "Volcanic Tuff as Secondary Raw Material in the Production of Clay Bricks"

_materials, 2021, doi:10.3390/ma14226872_

Round 1

Reviewer 1 Report

The article under review deals with the use of grinded zeolitic tuff from Romania as an additive in the preparation of clay bricks. It is a very interesting approach which could lead to the reduction of CO2 emissions in the general outcome. However, the manuscript has serious flaws, which, I believe, inhibit its consideration for being publishes in the "Materials" journal. The main drawbacks of the manuscript are detailed bellow, whereas further details are highlighted in the pdf document attached.

  1. It should be clearer that the samples selected for the present work belong to the “Dej Tuff” geological formation. A more thorough description of this formation should be given, detailing the sampling location. A simplified geological map of the area will benefit the manuscript.
  2. The section “Investigation methods” should be better renamed to “Analytical Methods” and should include and document all the methods employed. XRD and shrinkage estimation are completely absent and should be added. Details of the instruments used are fragmentary and should include type of the instrument, analytical conditions, details on sample preparation and laboratory affiliation.
  3. The original raw materials are partially described. Details concerning mineralogy and color should be given.
  4. The results should be presented in greater detail. The X-ray diffractograms shown are either of bad quality or partial (the diffractograms of the 1000 oC and 1100 oC should be given and commented). The absolute (numerical) values of density, shrinkage and compressive strength should be given in a separate table and should be discussed in more extend. Sintering/vitrification stage should be approached through SEM analysis performed on fresh fractures of the experimental firings of the various mixtures.
  5. The section about the compessive strength should be re-written, mainly because of the poor quality of the English language used but also because it includes several speculations which are not documented with the results obtained.
  6. The English language in general should be checked by a native language speaker. In many instances it is difficult to follow the text because of language inconsistencies.
  7. The "conclusions" section contains many repetitions as well as inferences which are not deduced from the results presented.

For all the above mentioned reasons I believe that the results presented are from a very preliminary stage from an interesting however work and need to be further enriched and elaborated in order to improve the manuscript in a way to be possible to consider it for publication in the “Materials” journal.

Author Response

Dear Reviewer

We want to thank you for the careful reading of the article and for their valuable suggestions.

According to your recommendation, we have corrected some paragraphs of the article and the other has been completed so that the results to be better highlighted.

The article under review deals with the use of grinded zeolitic tuff from Romania as an additive in the preparation of clay bricks. It is a very interesting approach which could lead to the reduction of CO2 emissions in the general outcome. However, the manuscript has serious flaws, which, I believe, inhibit its consideration for being publishes in the "Materials" journal. The main drawbacks of the manuscript are detailed bellow, whereas further details are highlighted in the pdf document attached.

  • It should be clearer that the samples selected for the present work belong to the “Dej Tuff” geological formation. A more thorough description of this formation should be given, detailing the sampling location. A simplified geological map of the area will benefit the manuscript.

Answer: Thank to the reviewer for the suggestion. More detailed description about the geological formation of used tuff was not intended in the present work since it was done earlier by the other author as in ref. 8. We added the area of the origin of used tuff (line 94).

  • The section “Investigation methods” should be better renamed to “Analytical Methods” and should include and document all the methods employed. XRD and shrinkage estimation are completely absent and should be added. Details of the instruments used are fragmentary and should include type of the instrument, analytical conditions, details on sample preparation and laboratory affiliation.

Answer: Thank you for these suggestions. The “Investigation methods” has been renamed as “Analytical Methods” (line 111). The analytical methods section was amended as suggested (lines 114, 119; 120-127)

  • The original raw materials are partially described. Details concerning mineralogy and color should be given.

Answer: The description of the raw materials was updated as requested.

  • The results should be presented in greater detail. The X-ray diffractograms shown are either of bad quality or partial (the diffractograms of the 1000 oC and 1100 oC should be given and commented). The absolute (numerical) values of density, shrinkage and compressive strength should be given in a separate table and should be discussed in more extend. Sintering/vitrification stage should be approached through SEM analysis performed on fresh fractures of the experimental firings of the various mixtures.

Answer: The results was updated as requested. The X-ray of 1000֯C is almost identical with the X-ray of 900֯C, therefore was not presented.

  • The section about the compressive strength should be re-written, mainly because of the poor quality of the English language used but also because it includes several speculations which are not documented with the results obtained.

Answer: The section was updated as requested. Thank you for recommendations.

  1. The English language in general should be checked by a native language speaker. In many instances it is difficult to follow the text because of language inconsistencies.

Answer: The language was improved. Thank you for recommendations.

  1. The "conclusions" section contains many repetitions as well as inferences which are not deduced from the results presented.

Answer: The conclusion section was updated. Thank you for recommendations.

For all the above-mentioned reasons I believe that the results presented are from a very preliminary stage from an interesting however work and need to be further enriched and elaborated in order to improve the manuscript in a way to be possible to consider it for publication in the “Materials” journal.

Reviewer 2 Report

I suggest that if the paper improved after revision it may be of interest for international community. One main drawback is that authors used lousy terms (from geological point of view). Some important data on mineralogy of rocks and their water content are absent. Most minor remarks may be easily corrected. 

Remarks and questions

Line 73. I suggest that it must be acid tuff or rhyolitic tuff.

Line 75. The term chalco-alkaline tuff is completely not correct. Is the meaning of chalco coper-bearing? How alkaline tuff is related to former indicated acid tuff? What is the meaning of term?

Line 76-77. It least the reference on zeolite minerals is needed. It is the single place in the paper where zeolite minerals are mentioned. What is the zeolite wt.% in the zeolitic tuff?

Line 77. One term, fine tuff, indicates the grain size, another term, marly, likely referred to its clay composition. Thus, the statement of alternation of fine and marly layers is not correct.

Fig. 2b. The scale is not shown.

Fig. 2d. Below the glassy character of tuff was indicated. But, the XRD pattern does not indicate the glassy, amorphous character of tuff. This is the single information that can be obtained from presented XRD picture. Related to the presented in the report data the picture is not informative.

Table 1. It is important. But the table must be corrected for to be of interest for international community. Namely, the mineral composition of yellow, and grey clays and tuff must be included. Water content of all rock types must be included. Wt.% of zeolites mast be included.

Table 2. How this table is correlated with Fig 2a,b,c? For example, in Fig 2a most grains are bigger than 63 microns?

Line 134 indicates that the authors are talking about vitric tuff (90% of glass). In this case how this rock was termed zeolitic tuff?

Lines 156-157. What kind of clay/tuff mixtures represent DTA-TG curves?

Figure 4a. Why the peaks related to clay minerals are absent?

What is the meaning of 5% etc?

Former, line 95, the mixing has different values.

20% - if it is highest tuff/clay rich mixture, why the intensity of quartz reflection is lower?

Line 166. That T range may be related to decomposition of clays and/or zeolites.

Lines 194-195. Very strange statement. The almost complete dehydration of zeolites occurs at T about 200 C. The same is pertinent to line 197.

By the way, where is the peak location on the figure 4b?

Lines 236-238. May be because that temperatures are not high enough for melting?

Table 3. Because of not clear terms used in the paper, what is the meaning of natural zeolite and zeolite rock used in the table?

Line 296. What is the meaning of mechanical strength or rock? Such term is used for the fist time in the paper, and it is used in the conclusion.

The paper is full of minor mistakes like cm3 (without superscript, e.g. line 190) and temperature, written like 1000º (line 222). In the table 3, wt.% is completely not correctly written.

Author Response

Dear Reviewer

We want to thank you for the careful reading of the article and for their valuable suggestions.

According to your recommendation, we have corrected some paragraphs of the article and the other has been completed so that the results to be better highlighted.

I suggest that if the paper improved after revision, it may be of interest for international community. One main drawback is that authors used lousy terms (from geological point of view). Some important data on mineralogy of rocks and their water content are absent. Most minor remarks may be easily corrected. 

Answer:  Thank you for these suggestions. The description of the raw materials was updated as requested (lines 97-101; Fig. 2)

Remarks and questions

Line 73. I suggest that it must be acid tuff or rhyolitic tuff.

Answer: Thank to the reviewer for the observation. We have corrected. 

Line 75. The term chalco-alkaline tuff is completely not correct. Is the meaning of chalco coper-bearing? How alkaline tuff is related to former indicated acid tuff? What is the meaning of term?

Answer: Thank to the reviewer for the observation. We have corrected. 

Line 76-77. It least the reference on zeolite minerals is needed. It is the single place in the paper where zeolite minerals are mentioned. What is the zeolite wt.% in the zeolitic tuff?

Answer: Thank to the reviewer for the observation. The reference has been added.

Line 77. One term, fine tuff, indicates the grain size, another term, marly, likely referred to its clay composition. Thus, the statement of alternation of fine and marly layers is not correct.

Answer: Thank to the reviewer for the observation. We have corrected. 

Fig. 2b. The scale is not shown.

Answer: Thank to the reviewer for the observation. We have corrected. 

Fig. 2d. Below the glassy character of tuff was indicated. But, the XRD pattern does not indicate the glassy, amorphous character of tuff. This is the single information that can be obtained from presented XRD picture. Related to the presented in the report data the picture is not informative.

Answer:  We change the XRD, it is present the wide hallo specific for amorphous materials.

Table 1. It is important. But the table must be corrected for to be of interest for international community. Namely, the mineral composition of yellow, and grey clays and tuff must be included. Water content of all rock types must be included. Wt.% of zeolites mast be included.

Answer:  The mineralogical composition was presented in previous paper as cited as ref. 25.

Table 2. How this table is correlated with Fig 2a,b,c? For example, in Fig 2a most grains are bigger than 63 microns?

Answer: As specified in line 129-132 the particles are agglomerated which are visible in Figure 2 a,b,c

Line 134 indicates that the authors are talking about vitric tuff (90% of glass). In this case how this rock was termed zeolitic tuff?

Answer: Thank to the reviewer for the observation. We have corrected. 

Lines 156-157. What kind of clay/tuff mixtures represent DTA-TG curves?

Answer: We have updated. The DTA-TG curves – sample 10% tuff (line 183)

Answer: Thank to the reviewer for the observation. We have updated in the text. 

Figure 4a. Why the peaks related to clay minerals are absent?

Answer: We have updated in the text. 

What is the meaning of 5% etc?

Answer: We have updated. It was typing error. Thank for the observation.

Former, line 95, the mixing has different values.

Answer: Thank to the reviewer for the observation. We have corrected. 

20% - if it is highest tuff/clay rich mixture, why the intensity of quartz reflection is lower?

Answer: The scale is different.

Line 166. That T range may be related to decomposition of clays and/or zeolites.

Answer: Thank to the reviewer for the observation. We have corrected. 

Lines 194-195. Very strange statement. The almost complete dehydration of zeolites occurs at T about 200 C. The same is pertinent to line 197.

Answer: Thank to the reviewer for the observation. We have corrected. 

By the way, where is the peak location on the figure 4b?

shown in the attached figure

Lines 236-238. May be because that temperatures are not high enough for melting?

Answer: The observation is True. The whole idea of sintering is to avoid the melting of the samples, to reduce the shrinkage and to have good mechanical properties. This approach is well known in powder metallurgy.

Table 3. Because of not clear terms used in the paper, what is the meaning of natural zeolite and zeolite rock used in the table?

Answer: They are identically, modified.

Line 296. What is the meaning of mechanical strength or rock? Such term is used for the fist time in the paper, and it is used in the conclusion.

Answer: They are identically, modified.

The paper is full of minor mistakes like cm3 (without superscript, e.g. line 190) and temperature, written like 1000º (line 222).

Answer: Thank to the reviewer for the observation. We have corrected. 

In the table 3, wt.% is completely not correctly written.

Answer: Thank to the reviewer for the observation. We have corrected. 

Round 2

Reviewer 1 Report

The revised ms was substantially improved. However the authors have only partially replied to the comments of the first revision or made the requested revisions. This is evident event in one of the pdf files attached (materials-1367783-peer-review-v1_REV_remaining points.pdf) which is cleared from those points accomplished by the authors. A great part of the comments still remains to be answered or corrected.

Some fewer comments are also included in the updated version of the ms, with regards to the newly inserted SEM micrographs (see the second pdf attached: materials-1367783-peer-review-v2_REVIEWER). These certainly improve the overall merit of the ms, however the authors still need to discuss them in the text (and/or in the caption).

My final concern is about the English language. The authors made a good effort of making some corrections, however I believe that the ms needs to be further corrected. This is more evident in the section relative to the compressive strength. To conclude and considering that I am not a native speaker, I would suggest that the editor decides on this matter.

Finally, I believe that if the proposed corrections/comments are corrected/taken into consideration (or if not then to have an appropriate justification) then the manuscript could be taken in consideration for being published in Materials journal.

Author Response

Rev 1 remaining points

All language corrections suggested in the text were made.

Line 71 references were added.

Line 102 modified to the final temperature.

Line 107 scanning electron microscope.

Table 1 empty column removed.

Table 2 heading modified.

Figure 4 updated.

Sintering documents by SEM images also.

Color was described using the hexadecimal RGB system.

The paragraph regarding the compressive strength was rewritten.

Table 3 heading modified.

review-v2_REVIEWER

fig 2b um to µm corrected.

Fig 6 modified and discussed.

The paragraph regarding the compressive strength was rewritten.

Reviewer 2 Report

The file is attached

Author Response

Reviewer: The new version of the paper still has the traces of the former version. Thus, the paper became less clear than before. For example: Line 51 – zeolitic tuff; line 69 – the addition of zeolitic tuff; line 74 Lines 78-81. There is a row of zeolite minerals. Why were these minerals reported? If they are
important, what zeolite wt.% in the tuff? The correct is analcime

Authors:  Corrected

Reviewer: Line 94 – Transylvanian Basin

Authors:  Corrected

Reviewer: Lines 98-99. It least between minerals commas are needed.

Authors:  corrected.

Reviewer: Sentences that are completely unclear: lines 136-137; 146-148; 195-197; 231-235; 237-239; 274-275; 294-295

Authors:  Lines corrected.

Reviewer: Figure 2 a,b,c. It has to be clearly written that the presented figures are photos of powdered
samples, not rock samples.

Authors:  Corrected.

Reviewer: Table 2. What is the meaning of D10, D50, and D90?

Authors: We defined the D10, D50, and D90.

Reviewer: Figure 3. What kind of mixture was pressurized? What value of pressure was used for mixed clay-tuff samples?

Authors:  Corrected, it was 40 MPa

Reviewer:  Lines 223-225. The reference is needed for the statement of zeolites sintering properties.
Lines 242-243. Where is the graph? The indications of sintering and more intense sintering have to
be indicated in the pictures.

Authors:  corrected.

Reviewer: Figures 5, 7, and 8. I am suggesting that the correlation between the data on density, shrinkage, and compressive strength must exist. Thus, the data presented on shrinkage are completely unclear. Such data have to be rechecked or, if they are true, describe particularly. Why, the reduction of
shrinkage is so prominent and out of trend for mixture with 5 wt.% of tuff?

Authors:  rechecked, corrected. There was a mistype (‘-‘).

Reviewer: Table 4. Has to be properly arranged.

Authors:  Corrected.

Reviewer: Axis y – compressive strength

Authors:  Corrected.

Reviewer: Table 6. The present version is completely unclear. What is the meaning of column (%)? To what kind of rock or mixtures the data of compressive strength correspond? How related zeolite wt. % to the present study? Why there are not any data on ignimbrite tuff/clay mixtures?

Authors:  Table 6 modified, text updated.

Round 3

Reviewer 2 Report

the file is attached

Author Response

Reviewer: I do not understand the following sentences:

Lines: 70-71, 137-139, 149-150, 153-154, 167-169, 175, 206-207, 229-232, 266-267, 275-289, 317-319

Author:

70-71 modified,

137-139 table 1 modified to present the elemental analysis not the oxide composition

 149-150 modified to particle size analysis

 153-154, modified. This is important for the green density and sintering behavior

167-169 It is unclear to us what is difficult to understand in this sentence. In it’s natural form the clay particles agglomerates, therefore, in the SEM images they look like they are coarser particles. The other part of the sentence refers to the fact that in the green state of the samples (manufactured according to the described methodology) our samples can be safely handled without the danger of damaging them. If you can suggest a better formulation, we are welcoming it.

 175 modified

 206-207 modified

229-232 modified

266-267 modified

275-289 modified

 317-319 modified

Reviewer: Line 204: correct the writing of were

Author: modified.

Reviewer: Table 1. All data are shown on anhydrous basis. It must be included. Table 1 modified

Author: modified. The table 1 content was modified to contain the results of the primary EDS analysis

Reviewer: Line 172-175. I do not understand the figures related to XRD reflections.

Author: Modified. If it is further unclear, please be more specific.

Reviewer: Figure 4c. The meaning of t°C and m wt. % must be included into the caption.

Author: Added

Reviewer: Line 240. I suggest that the correct ref is on fig. 6. 

Author: The text in line 240 refers to figure 7 The sintering behaviors of our samples was discussed in the light of the SEM images and completed with the shrinkage behaviors. To avoid misunderstandings, the line was repositioned after the figure 6.

Reviewer: Lines 301-308. It seems to me that quartz is the main component participated in the melting reactions. Thus, the lowering of compressive strength has to be related in something else.

Author: Since all samples had high quartz content at least part of it remains unreacted during sintering. These quartz particles on cooling will form cracks (will not be strongly bonded to the surrounding matrix and will not strengthen the sample as a whole). This combined with the increased porosity are responsible with the strength decrease.

Reviewer: Lines 325-339. Common heat capacity of clays is 878 J/(kg K). Whereas tuff heat capacity is 1.165-1.305 J/(kg K). Thus, the paragraph following then conclusions are out of sense if the real heat capacity of components without consideration.

Author: Dear reviewer we redid the calculation using the suggested values and updated the manuscript. Let’s take into account an ordinary brick (240x115x63 mm). If this brick has a density of 1.71 g/cm3 (as in our control sample fired at 900 °C with no tuff) it has a mass of 2973 grams. A brick containing 30% tuff has a density of 1.52 g/cm3 and it has a mass of 2643. Being lighter it needs less heat to increase it’s temperature, however its temperature needs to be increased to 1100°C. Using a heat capacity of 1165 J/kgK it needs 2802637.2 J compared to 2349265 J for the control sample to be heated to 900 °C, that is a 16% heat input increase. In the case of sample S3 the needed energy is 3034997 J and corresponds to an increase of over 22%.